# Recommendations for Identifying Valid Wear for Consumer-Level Wrist-Worn Activity Trackers and Acceptability of Extended Device Deployment in Children

**DOI:** 10.3390/s22239189

**Published:** 2022-11-26

**Authors:** David Wing, Job G. Godino, Fiona C. Baker, Rongguang Yang, Guillaume Chevance, Wesley K. Thompson, Chase Reuter, Hauke Bartsch, Aimee Wilbur, Lisa K. Straub, Norma Castro, Michael Higgins, Ian M. Colrain, Massimiliano de Zambotti, Natasha E. Wade, Krista M. Lisdahl, Lindsay M. Squeglia, Joseph Ortigara, Bernard Fuemmeler, Kevin Patrick, Michael J. Mason, Susan F. Tapert, Kara S. Bagot

**Affiliations:** 1Herbert Wertheim School of Public Health and Human Longevity Science, University of California, San Diego, CA 92093, USA; 2Center for Health Sciences, SRI International, Menlo Park, CA 94025, USA; 3Department of Radiology, University of California, San Diego, CA 92093, USA; 4Instituto de Salud Global de Barcelona, 08036 Barcelona, Spain; 5Laureate Institute for Brain Research, Tulsa, OK 74136, USA; 6Department of Computer Science, University of Bergen, 5007 Bergen, Norway; 7Department of Family Medicine and Population Health, Virginia Commonwealth University, Richmond, VA 23284, USA; 8Department of Psychiatry, University of California, San Diego, CA 92093, USA; 9Department of Psychology, University of Wisconsin-Milwaukee, Milwaukee, WI 53211, USA; 10Department of Psychiatry and Behavioral Sciences, Medical University of South Carolina, Charleston, SC 29208, USA; 11Center for Behavioral Health Research, University of Tennessee, Knoxville, TN 37996, USA; 12Department of Psychiatry, Icahn School of Medicine at Mount Sinai, New York, NY 10029, USA

**Keywords:** consumer wearables, physical activity, children, Fitbit

## Abstract

Background: Self-reported physical activity is often inaccurate. Wearable devices utilizing multiple sensors are now widespread. The aim of this study was to determine acceptability of Fitbit Charge HR for children and their families, and to determine best practices for processing its objective data. Methods: Data were collected via Fitbit Charge HR continuously over the course of 3 weeks. Questionnaires were given to each child and their parent/guardian to determine the perceived usability of the device. Patterns of data were evaluated and best practice inclusion criteria recommended. Results: Best practices were established to extract, filter, and process data to evaluate device wear, r and establish minimum wear time to evaluate behavioral patterns. This resulted in usable data available from 137 (89%) of the sample. Conclusions: Activity trackers are highly acceptable in the target population and can provide objective data over longer periods of wear. Best practice inclusion protocols that reflect physical activity in youth are provided.

## 1. Introduction

Physical activity is an important determinant of health in children and adolescents. Insufficient physical activity, particularly limited moderate to vigorous intensity activity (MVPA), is associated with obesity development of cardiovascular risks such as high levels of low-density lipoprotein cholesterol, elevated blood pressure [1], and progression of chronic diseases such as type 2 diabetes mellitus [2]. These conditions put children and adolescents at additional risk for long-term morbidity and mortality [3,4]. Physical activity and mental health are related, as children with low levels of physical activity exhibit greater social dysfunction [5,6], anxiety [7], depression [7,8] negative affect [9], stress response [10], poorer self-image and self-esteem [11], and overall diminished psychological wellbeing [12,13]. Conversely, the literature suggests that adolescents who engage in high levels of physical activity have better family and peer relationships [6], engage in more pro-social activities, exhibit less substance use, and demonstrate better academic performance than less active youth [14].

Much of the available data on relationships between physical activity, health, and development in child and adolescent populations have been cross-sectional and based on self-report or are drawn from laboratory-based results of physical capacity (vs. volume of physical activity). Other data are drawn from relatively short term (generally <8 days) remote monitoring using accelerometers which may miscategorize some common activities (i.e., cycling, weight lifting, swimming) and fail to recognize physical activity of very short duration (<1 min) [15]. Further, accelerometry based tools for longer-term remote monitoring have limited acceptability [16] for use by the general public, have data that are challenging to integrate across device manufacturers, and do not integrate multiple sensors for their determination of physical activity [17].

In the past decade, advances in consumer-level activity trackers allow for multisensory data collection typically using wrist-worn devices. These devices integrate, at minimum, accelerometers and photoplethysmography sensors (from which heart rate can be derived), provide continuous measurement at >1 Hz for extended periods of time. Additionally, these devices are widespread throughout the United States, with an estimated yearly market value of $75 billion by 2025 [18]. Such devices have recently been tested in children within our target age group providing relatively accurate performance for objective measures of physical activity intensity and volume across different activities [19]. Additionally, at least some commercially available wearable devices (most notably, Fitbit brand devices) have reasonably good agreement to previously utilized activity monitoring devices like the Actigraph GT3X+ in both adult and child populations. As such, they may generate data useful for comparison to previous studies and/or longitudinal studies that have previously deployed traditional accelerometers. However, unlike traditional actigraphy, to our knowledge there is not yet a best practice method to determine if the device is actually being worn, and criteria for inclusion/exclusion at the minute, daily, and weekly level have not been established. The primary aim of this project was to establish a best practice protocol to identify valid device wear and determine rules to ensure that included data represent typical patterns of activity and behavior.

While accuracy and precision are important to successful remote evaluation, participant perception of the device’s usability will likely determine long term adherence. Perceived usability of Fitbit devices in particular have been established in several adult groups including men with prostate cancer [20], endometrial cancer survivors [21], and individual’s with Type II Diabetes [22]. Additionally, Australian adolescents reported high levels of usability with the Fitbit flex [23], and parents of children ages 9–12 reported high acceptability of a similar wrist worn device (KidFit) [24]. We sought to expand on our understanding of the usability and acceptability of these devices in a younger (9–10 years old) population of children and their parents.

## 2. Materials and Methods

Participants: Data collection occurred at three of The Adolescent Brain and Cognitive Development (ABCD) 21 study sites with approximately 50 ABCD participants recruited per site. In addition to exclusionary criteria for ABCD described elsewhere [25] children were excluded from this substudy if: (1) if the primary parent/guardian did not own a Fitbit-compatible smartphone with which to sync the Fitbit device, (2) the parent was not fluent in English, or (3) the youth was not interested in wearing the Fitbit device continuously for 3 weeks. All children were asked to provide informed assent, and at least one of the parents provided informed consent. Both children and parents who assented/consented were given a description of the protocol in accordance with UCSD Human Research Protections Program.

Procedures: Fitbit devices were deployed for a period of 22 days (day of deployment + 3 weeks). For interested families, informed consent was completed with the parent and informed assent with the child, a Fitbit HR Charge (Fitbit Inc., San Francisco, CA, USA) was provided, and a study specific de-identified Fitbit account was established (using a participant ID number instead of name). Height, weight, sex, and handedness were recorded accurately, but age for all participants was entered as 13 years, as the Fitbit platform was unable to accept the study’s actual age range at the time (9–10 years). Research staff assisted parents in downloading the Fitbit app to their smartphones (the child’s phone was used in the rare instances that they had their own) and in pairing and syncing the Fitbit device to the phone. There were no GPS or notification functionality in the Fitbit device itself, but research assistants confirmed that notification, geolocation, and sharing capabilities were turned off to the associated smartphone.

The Fitbit accounts were linked to Fitabase (Small Steps Labs LLC, San Diego, CA, USA), a third-party research platform that aggregates Fitbit data with fine temporal resolution useful for research (e.g., 5–15 s epochs for heart rate, 30 s epochs for sleep, 1 min epochs for other variables). Both child and parent were provided with instructions for charging and syncing the device over the 3-week period. Data were securely synced on an ongoing basis to Fitabase, then securely imported to ABCD servers. Research staff manually entered wear dates into Fitabase and REDCap databases to ensure data integrity, as each Fitbit device was re-used.

Throughout the wear period, research staff logged onto a Fitabase dashboard daily to ensure ongoing wear and successful syncing of each active device. When new data were not displayed for three consecutive days, the family was called and sent text messages asking them to manually sync the device, reminding the participant to wear the device as much as possible, and to troubleshoot questions or concerns with the family.

Demographics. Key demographic variables [26] were obtained from the assessments collected during the standard ABCD baseline assessment from both the youth and participating parent. Height and weight were measured in person at the time of device deployment from which body mass index was calculated and reported as sex and age-specific percentiles [27].

Acceptability and Usability Scales: Questionnaires examining the usability and acceptability of the device and associated application were administered to the participating children both pre and post deployment to assess differences between expectations and experience associated with device usage and the feasibility of long-term deployment. Parents were also queried regarding their experience during the deployment to better understand the feasibility of using this device for long term assessment in pre-adolescent populations.

Physical Activity Variables: The following variables were extracted from Fitabase at the minute level and used for analysis: number of steps, heart rate (HR), and activity intensity classification (i.e., sedentary, light activity, fairly active/moderate, and active/vigorous using CDC based MET classifications) [28]. Time spent in moderate-vigorous activity was combined to create a variable defined as ‘moderate to vigorous physical activity’ (MVPA). These data are available in minimally processed form at https://nda.nih.gov/abcd/query/abcd-curated-annual-release-3.0.html.

Data Processing: Data were processed using Python software. We established a protocol to assess valid wear data that carefully evaluated for patterns of missingness and aphysiologic values. First, data were assessed for missing HR values, as “baseline” values of 0 (steps),1.0 (METs), and 1 (sedentary in activity classification) are automatically generated for any minute when HR is not observed. When HR was missing, steps, METS, and intensity level variables were further analyzed for any non-baseline values. Minutes with missing HR and baseline values for other variables were classified as non-wear. Minutes with missing HR, but with higher than baseline values for non-HR based variables were assigned a HR value based on the average of the surrounding minutes (i.e one minute before, and one after the missing value(s)).

Although the Fitbit device does some amount of “prescreening” to provide heart rate data believed to be accurate, according to the device manufacturer, the accuracy of HR measurement can be affected by the specific location of the device placement on one’s wrist as well as the level of contact with skin (i.e., being too loose or too tight). With this in mind, time series data were further screened to determine the prevalence of unlikely/aphysiologic values at HR values below 40, 50, and 60 and above 200 to determine both the number of occurrences and total number of minutes implicated.

The HR monitors associated with Fitbit devices can be “turned on” by being placed too closely to a reflective or vibrating surface. In most cases this results in repeated HR values, potentially separated by periods of missing HR data. Because of this fact, along with the physiological the understanding that heart rate is dynamic with small fluctuations in value even at rest and during steady state activity, the data were further explored to identify strings of repeated values. This is in line with NIH recommendations regarding handling accelerometer-based data by identifying and excluding long runs of repeated values [29]. In this case, a repeat was defined as either an identical HR across sequential minutes, or a HR value repeated at either end of a string of missing HR values. For missing values to contribute to a repeat string, an identical HR value had to both precede and immediately follow the string of missing values. The number of instances and number of minutes affected were generated for repeated strings of 6+, 11+, 16+, 31+, and 61+ min.

After excluding minutes without HR, minutes with HR < 50 or >200 bpm, and minutes that were part of repeated strings of 11+ minutes, daytime and nighttime wear were decoupled. For the current analysis, time identified by the device’s internal algorithms as sleep was removed from consideration and the number of days with ≥600, ≥750 and ≥900 min of valid non-sleep wear were calculated to establish the minimum threshold of daily wear time to be considered representative of typical activity.

Based upon a threshold of ≥600 or more minutes of valid wear, valid days were further aggregated to the week level for: (1) all valid days, (2) weekdays only, and (3) weekends only, with Day 2 (the first full day following deployment) through Day 8 constituting Week 1, and Days 9–15 and 16–22 constituting Weeks 2 and 3, respectively. Cut-offs commonly deployed for accelerometer-based classification of activity were applied to determine levels of inclusion for any particular week [30,31,32]. Specifically, requirements for ≥3 valid days, ≥4 valid days, and ≥5 valid days per week with and without the requirement of a weekend day were evaluated.

Statistical Comparisons: Data were managed, analyzed and visualized using Microsoft Excel and STATA. Between subject comparisons utilizing unpaired *t*-tests were calculated to compare activity levels of boys vs. girls and paired *t*-tests were used to compare activity on weekdays vs. weekend days. Given the pilot nature of this study, and the primary goal of establishing a procedure for determining inclusion rules for Fitbit gathered data findings with *p* < 0.05 were interpreted as statistically significant.

## 3. Results

Over 7 months in 2017, 154 participants completed the protocol. Data from seven participants was lost due to insufficient device syncing or corrupted data transfer. Data from an additional eight participants had no valid heart rate values across the entire data collection period. We will explore our inclusion rationale in more detail below, but data from an additional two participants was excluded because there was not at least 600 min/day of daytime wear across at least 4 days (including a weekend day) for at least one of the three weeks of wear Table 1 provides key physical and socioeconomic demographics for the final sample of 137 participants.

The sample of nine and ten year old’s was 53% female and from a diverse sociodemographic and racial/ethnic background. No sex differences were seen for age, BMI, household income, or parent education (all *p* > 0.05).

### 3.1. Acceptability, Behavioral Reactivity, and Compliance

As shown in Table 2, most all of children were comfortable with (98%) and enjoyed wearing (87%) the Fitbit device over the deployment period. Further, the vast majority of youth (98%) and parents (92%) would be interested in/allow wear for a longer period of time. Real time and historical data from the device appear to have been regularly viewed with 74% of children reporting checking the device several times per day for updates on their activity, and 80% of children and 86% of parents reporting using the app or website to see activity information during the protocol period. Additionally, 42% of parents reported encouraging their child to change their activity based on the observed data, and 59% of parents and 48% of children believed that activity patterns were changed in response to the information received from the device and/or app. When considering compliance with 24 h wear protocols, 62% of youth reported removing the Fitbit daily (slightly less than the 67% who predicted they would remove the device at baseline), and 29% reported sometimes forgetting to put it back on after taking it off.

### 3.2. Determining Wear Time

The number of minutes excluded, and number of participants affected, at each of the heart rate levels explored as being outside of normal ranges for children [33] are shown in Table 3.

Only two children showed heart rates above 200 bpm, and each of those children only had one minute with this value. Given that, in both cases, this value was not surrounded by similarly high heart rate values (i.e., in the 190’s) these were thought to be likely aphysiological noise, and the group proceeded recommending that heart rate values >200 be considered non-wear minutes in future analysis by ABCD investigators. When exploring the lower bounds of physiological likelihood, 2 children (1%) had HR values lower than 40 bpm and 23 (17%) had HR values lower than 50 bpm. The total number of minutes that fell below these values were 65 and 10,676, respectively (<0.01% and 0.3%, respectively of total minutes with heart rate). In contrast, 124 (89%) children had at least one minute with a heart rate below 60 bpm with a total of 124,955 (4.1%) of minutes with HR falling below this level. Although bradycardia below 50 bpm is not impossible, it was determined by the study team that given the relatively low volume of these events, and the high likelihood that they were artifact caused by incorrect wear, values below 50 bpm should be considered non-wear minutes.

The number of minutes included in a repeat string based upon different length criteria is shown in Table 4.

As expected, reducing the number of minutes required to indicate a repeated string increased the total number of excluded minutes, and the number of instances that repeated strings were observed per participant. With 6+ minutes, there was an average of 61.5 instances accounting for 4282 min per child across the three-week wear period. The number of instances dropped dramatically to 21.2 per child when the criteria to initiate a “repeat” string was increased to 11+, although the affected number of minutes only decreased slightly more than 5% to 3999. Similar trends of relatively large decreases in incidence, but modest changes in the number of minutes implicated are observed when the criteria to initiate is increased to 16+, 31+ or 61+ minutes. This indicates that the vast majority of the affected minutes are part of very long strings (i.e., >60) of unvaried repeated values. In an effort to balance physiological likelihood with preserving the largest volume of data possible, the investigative team determined that strings greater than 10 min should be considered non-wear and excluded from analysis.

### 3.3. Wear Time Thresholds for Day Level Inclusion

The number of days included per participant using different thresholds of valid (daytime) wear minutes are shown in Table 5.

When using a threshold of ≥600 min/day for inclusion participants achieved an average of 15.2 valid days (73% of total possible days). When thresholds were increased to ≥750 and ≥900 min/day number of valid days per participant declined to 12.2 (58%) and 4.4 (21%), respectively. To preserve a relatively large volume of data, and to match data inclusion rules utilized by other/historical methods of remote monitoring like accelerometry [34] the investigative team agreed that ≥600 min/day would be used to define a valid day that is indicative of whole day activity.

### 3.4. Wear Time Thresholds for Weekly Level Inclusion

The protocol deployment period in the current study was substantially longer than the standard seven to ten days utilized for other methods of remote monitoring [35,36]. With this in mind, the team explored the number of weeks each participant had that were indicative of regular activity (i.e., valid) using different numbers of valid days as total days, and also separated by weekday and weekend day. These data are shown in Table 6.

As expected, fewer weeks are included with increasing requirements, particularly when at least one weekend day must be included. However, the changes in number of included weeks in not linear, with a larger drop in the percentage of weeks included when inclusion rules move from 5 to 4 days (72% vs. 82%) then when inclusion rules move from 4 to 3 days (82% vs. 87%). Data regarding daily number of steps and minutes in moderate-to-vigorous activity provided in Appendix A indicate a difference in activity on weekday vs. weekend days (*p* < 0.0001 for both). Based on these data, combined with earlier accelerometry based recommendations suggesting that four days with at least one being a weekend is sufficient to exemplify “typical” activity [32], the investigative team adopted guidelines of 4+ days with at least one being a weekend day to be included as a valid week. A summary of all best practice recommendation steps with rationale are listed below in Table 7.

### 3.5. Steps, Resting Heart Rate, and Minutes of Activity across Intensity Levels

Using the data inclusion steps outlined above, data were evaluated to determine activity levels in this population, as well as to examine differences by sex and between weekday and weekend day activity. These data are presented in more detail in tabular form in our Appendix A. In short, girls wore the device less than boys (*p* = 0.027), had higher resting heart rates (*p* < 0.003) and appear to have engaged in less activity collecting both less steps (*p* < 0.001) and minutes of MVPA per day (*p* <0.001). Across the entire sample, wear time was slightly higher on weekdays compared to weekends (*p* = 0.012). Further, as previously noted, all participants achieved a higher volume of physical activity on weekdays vs. weekend days when measured by either steps or minutes of MVPA (*p* = < 0.0001 in both cases). In our Appendix A we also include some data regarding mostly non-significant changes in wear time and physical activity behaviors compared across the three weeks of wear.

## 4. Discussion

Recent studies have demonstrated the validity of wearable activity trackers, in particular Fitbit, relative to gold standard laboratory measures, in children [19,37,38]. Our findings help to establish the feasibility of deploying these trackers in large populations over extended periods of time. Specifically, we observed that Fitbit Charge HR devices are well accepted by both young children (9–10 years old) and their parents, and that these devices are worn sufficiently often/long to capture substantial amounts of typical physical activity behavior in the majority of children across an extended wear period. These findings are consistent with previous research demonstrating relatively high levels of acceptability of wearable trackers in older children and adolescents [23,24].

Further, using these data, we developed processes for examining data from Fitbit to determine valid wear. We believe that these protocols are in line with previous methods of determining valid wear with accelerometers [29], and should yield the best blend of physiological likelihood, identifying actual human wear, and data inclusion. These methods will be utilized by ABCD investigators in the future, and we believe that these, or similar methods should be adopted as best practice for working with data from wearable devices that integrate accelerometer and polyplesmography signals.

These data highlight the enormous potential for using commercially available wearable monitors for continuous objective measurement of physical activity, sedentary behaviors, and sleep among youth using ubiquitous technologies that are designed for extended wear. While there was some loss of data in this study due to factors such as insufficient wear, or lack of data syncing from the device, usable physical activity data were collected from the majority of the sample (≥4 days/week and ≥1 weekend day for 89% of the sample), levels similar to those shown by NHANES and other large-scale studies using objective measures [29]. Further, while we recommend validating data at the weekly level to ensure that captured data represents typical behavior and is comparable to earlier device-based observations, we also recognize that there may be value in including the larger number of valid days made possible through acceptable extended wear (>15/participant average), regardless of the week of capture across the extended wear period.

Finally, analysis of the dataset generated following these procedures indicated differences in activity between boys and girls at this age group, and differences in activity measured via steps and minutes of MVPA between weekdays and weekends. This adds to evidence that suggests that interventions designed to increase activity on the weekend in children, and girl children in particular, may offer meaningful public health opportunities.

Due to differences in movement patterns, and the possibility of large amounts of movement in the limbs unassociated with substantial energy expenditure (as might occur during a particularly engaging bout of video game playing) there is difficulty in determining energy expenditure and physical activity categories from wrist-based devices. Multiple sensors promise to address this difficulty by including heart rate, but photoplethysmography based assessment of heart rate has been reported to have poor accuracy in individuals with darker skins and appears to have reduced accuracy at higher heart ratees. As such, these devices may be limited in their ability to provide highly accurate assessment of energy expenditure in particular, and other heart rate associated data like sleep stage and heart rate variability across long periods of time, particularly in non-white individuals.

Additionally, the mobile technology field changes rapidly, which can be challenging within a longitudinal research environment. Indeed, by the time of completion of our laboratory-based validation study [29], and wide-spread implementation of device deployment in the larger ABCD study, an updated version of the validated device (Fitbit Charge HR) was released (the Fitbit Charge HR 2), and several iterations have been released since (as of the writing of this paper we are on the Fitbit Charge HR 5). Based upon reports from the manufacturer, the updated models do not differ in the manner in which the metrics reported here are generated, although improvement in sensors may result in improved accuracy from model to model, and new measurement features like VO2_max_, tracking of menstrual cycles, and detection of sleep apnea are now available. We have anticipated that updated versions of all technologies used in ABCD will be released during the lifespan of this longitudinal study but believe that the wear time rules established here are sufficiently robust to allow for cross-device comparisons. We do, however, recognize that this may be an ongoing limitation to the research, and a challenge across the research community. An additional limitation to these data is that we did not record environmental context during wear, so we are unable to draw conclusions about the reasons for (non) wear or the context of the observed activity.

Despite these limitations, these data indicate that we can gather extended data regarding physical activity, sleep, and sedentary behaviors reliably from participants using Fitbit based wearable technology in the ABCD study. Gathered data can then be coupled with biological, neurocognitive, psychosocial, and neuroimaging data, to link recent physical activity to psychological health and well-being. The opportunity for prospective and longitudinal study of these reciprocal relationships highlights the potential contribution of ABCD to our understanding of the bidirectional relationships between physical activity, health, and childhood development. Future studies in this area should examine youth acceptability of, and adherence to using, wearable activity trackers and the trajectory of physical activity over longer periods of time (months or years), including the impact of certain characteristics (e.g., baseline physical activity, engagement in sports, parent involvement/monitoring, socioeconomic status, sex and age) on device acceptability and wear time. Additional validation of newer models vs. older models within the same device maker, or the precision across different manufacturer’s devices would also be valuable to the field to establish the comparability of data collected longitudinally and across different studies. Finally, as additional sensors are integrated into commercially available devices research to establish best practice for including these data will be necessary.

## 5. Conclusions

Commercially available wearable devices that integrate multiple sensors can be effectively used in research among children. These data show that the Fitbit Charge HR is acceptable in youth populations, who would be willing to wear it for extended periods of time (>3 weeks at minimum.). Further, by providing processing rules that utilize granular data that can be preserved over time, we have helped to bridge the gaps in practice between traditional “research-grade” accelerometers and devices that are commercially produced. Future research using high resolution data from Fitbit devices can capitalize on these methods to draw valid inferential conclusions regarding a wide range of health-related research questions.

## Figures and Tables

**Table 1 sensors-22-09189-t001:** ABCD Fitbit substudy participant characteristics (N = 137).

	% or Mean (SD, Range)
% Female	53%
Age (years)	9.97 (0.60, 9–10.99)
Race/ethnicity:	
Asian	10%
Black	5%
Hispanic	22%
Multiracial	12%
Pacific Islander	1%
White	50%
Body Mass Index percentile by age	58.28 (32.86, 1st–99th)
Parent education:	
<High school Diploma	4%
HS Diploma/GED	4%
Some College	30%
Bachelor	33%
Post Graduate Degree	29%
Annual household income in USD:	
<$50 K/year	18%
$50–100 K/year	20%
>$100 K/year	51%
Don’t know/decline	11%

**Table 2 sensors-22-09189-t002:** Subjective experience: acceptability, behavioral reactivity, and compliance reports by Youth and Parent before and after the 3-week Fitbit wear period.

Youth Survey	Pre-Wear	Post-Wear
Thought wearing the Fitbit would be not at all or a little annoying	80%	-
Comfortable or very comfortable wearing a Fitbit in front of friends	-	98%
Thought it would take/it took a few hours or less to learn to use Fitbit	60%	65%
Thought would/did enjoy using the Fitbit a lot	86%	87%
Did not find the Fitbit to be too complicated	-	87%
If asked to wear the Fitbit for longer, would do so		98%
Thought would/did change activity a little or a lot while wearing Fitbit	50%	71%
Checked Fitbit several times per day or more for activity information	-	74%
Checked the Fitbit app or website for activity information	-	80%
Changed activity based on Fitbit app or website information	-	48%
Thought would have to/had to remove the Fitbit once per day	67%	62%
Took off Fitbit for:		
Bathing		92%
Sports		17%
Swimming		46%
Sleep		7%
Other (e.g., school)		20%
Sometimes forgot to put Fitbit back on after taking it off	-	29%
I found the Fitbit too complicated (%DISAGREE)	-	87%
I felt confident using the Fitbit (%AGREE)	-	92%
**Parent survey:**	**Pre-wear**	**Post-wear**
Would allow their child to wear the Fitbit for a longer period of time in the study	-	92%
Used the Fitbit app and/or website to see their child’s activity	-	86%
Encouraged their child to change their activity based on the Fitbit app and/or website information (of parents who used app/website)	-	42%
Their child used the app and/or website to see his/her Fitbit activity	-	68%
Reported child changed their activity based on the Fitbit app and/or website information (of youth who used app/website)	-	59%

**Table 3 sensors-22-09189-t003:** Data exclusion based on aphysiologic signal—unlikely heart rate values.

	Number of Minutes Affected	Number of Participants Affected	Range per Person	
	n (%)	n (%)	Low	High
Minutes with HR <40	64 (<0.01%)	2 (1%)	0	45
**Minutes with HR <50**	**10,676 (0.3%)**	**23 (17%)**	**0**	**3430**
Minutes with HR <60	124955 (4%)	124 (89%)	0	11469
**Minutes with HR >200**	**2 (<0.01%)**	**2 (1%)**	**0**	**1**

*Note*: bold rows indicate the choices made for data processing at daily level.

**Table 4 sensors-22-09189-t004:** Data exclusion based on aphysiologic signal—repeated heart rate values.

	Total Sample		Per Participant	
Repeat Length (min)	Instances	Minutes Excluded (%)	Instances Mean (±SD)	Minutes Excluded Mean (±SD)
6+	8549	595,219 (19.3%)	61.5 (31.7)	4282 (5791)
**11+**	**2942**	**555,847 (18.0%)**	**21.2 (14.0)**	**3999 (5842)**
16+	2148	545,841 (17.7%)	15.5 (11.0)	3926 (5850)
31+	1397	529,652 (17.2%)	10.1 (7.7)	3810 (5857)
61+	895	508,083 (16.5%)	6.4 (5.4)	3655 (5863)

*Note*: bold row indicates the choice made for data processing at daily level.

**Table 5 sensors-22-09189-t005:** Data inclusion based on daily wear time, per participant.

Inclusion Criteria	Valid Days across Protocol Period Mean (SD)	% of Total Possible Days
**≥600 min/day**	**15.2 (5.0)**	**73%**
≥750 min/day	12.2 (4.9)	58%
≥900 min/day	4.4 (3.3)	21%

*Note*: bold row indicates the choice made for data processing to daily level.

**Table 6 sensors-22-09189-t006:** Data inclusion based on weekly wear for the entire sample.

Assuming ≥600 Valid min for Each Day	Number of Participant Weeks (Total Possible = 417)	% of Total Possible Weeks
≥3 days/week	363	87%
≥3 days/week with ≥1 weekend day	356	85%
≥4 days/week	340	82%
**≥4 days/week with ≥1 weekend day**	**335**	**80%**
≥5 days/week	299	72%
≥5 days/week with ≥1 weekend day	295	71%

*Note*: bold row indicates the choice made for data processing to weekly level.

**Table 7 sensors-22-09189-t007:** Recommended best practice for evaluating Fitbit data to assess daily and weekly physical activity in youth.

Standard	Rationale
Exclude min with no heart rate value	Indicates non-wear (see Table 3)
Exclude min with heart rate values <50 or >200 bpm	Aphysiological and likely due to artifact (see Table 3)
Exclude min in which heart rate is repeated for 11+ minutes	Aphysiological and likely due to artifact (see Table 4)
Exclude days with <600 min of daytime wear	Unlikely to represent normal daily activity (see Table 5)
Exclude weeks with <4 days (1 of which must be a weekend)	Unlikely to represent normal weekly activity (see Table 6)

## Data Availability

Datasets for this study, and for the larger ABCD dataset are available upon request from UCSD based site PI’s. The ABCD data repository grows and changes over time. The ABCD data used in this report came from the ABCD 3.0 data release (https://doi.org/10.15154/1519007).

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
