# Peer review of "Recommendations for Identifying Valid Wear for Consumer-Level Wrist-Worn Activity Trackers and Acceptability of Extended Device Deployment in Children"

_sensors, 2022, doi:10.3390/s22239189_

Round 1

Reviewer 1 Report

Dear authors,

Review the document.

Thanks.

Author Response

This is an exact copy of the attached file, although the attachment includes our responses to the other reviewer as well.

Reviewer One

Major Heading

It's an interesting sub-study. It is hoped that they can materialize the projections of the research that they have declared, and that they can show us the study in its entirety and the results of those projections so that we are not left only with the "sub-study".

We thank the reviewer for their thorough reading and associated recommendations.  We have tried to make clear that this is one of many sub-studies associated with the Adolescent Brain and Cognitive Development (ABCD) study.  ABCD is designed as a 10 year, 22 site, longitudinal study that measures numerous physiological, psychological, and behavioral variables.  This “sub” study was designed as a means of determining if consumer wearables would be useful for long term deployment in a youth and adolescent population, and to establish best practice rules allowing for the analysis of comparitively granular data gathered at the minute level instead of relying only on daily totals of processed data provided by Fitbit.  In doing this, we also provided some data about the cross-sectional observations of the (relatively small) observed population.

Findings from this study have been implemented in the larger ABCD study leading to a much larger quantity and variety of data. Researchers who use the data from the larger effort will look to this study for methodological insight and will cite it in their work. 

  1. The abstract should not exceed 200 words

We have reduced the word count in the abstract to 153 words.

  1. The statistical program used must be mentioned (line 175, “Statistical Comparisons”)

We have clarified that we used Python to process data (line 194) and Microsoft Excel and State for data management, analyses, and visualizations (line 234).

  1. What do the authors think about the obsolescence of the data (Year 2017)? Is that why they called it a “sub-study” (line 181)

We appreciate the reviewer’s observation and recognize that there are rapid changes to the technological landscape in this area. This study was a natural extension of early work done by our group to assess the utility of Fitbits and/or other devices for deployment in research by assessing in-lab accuracy and precision compared to reference devices widely used in research. The publication from that study is entitled “Performance of a Commercial Multi-Sensor Wearable (Fitbit Charge HR) in Measuring Physical Activity and Sleep in Healthy Children.” It is important to note, that although the form factor and ways of deriving new metrics from consumer-level wearables change rapidly, the underlying technologies of PPG sensors and accelerometers have remained consistent for some time now. Additionally, the data on physical activity and sleep are a valid representation of the constructs no matter how much time has passed. 

This follow up study was done to assess the acceptability of deployment of Fitbits in the youth population. In essence, having shown reasonably high levels of accuracy and precision, ABCD wanted to ensure that Fitbit would be well received in this population, and that it could yield greater amounts of data, over a longer period of time, than traditional accelerometery.

Further, heretofore, commercial wearables have not had the same level of rigor applied to determination of what qualifies as “real” wear, leading to potential to both over- and under-estimate various types of behavior.  We believe that these analyses close that gap in practice by establishing rules for inclusion and will be helpful to researchers for years to come.

  1. The statistical analysis is very basic: only descriptive comparison and comparisons of means. Why not present the complete study? What do authors think?

We appreciate the reviewer’s observation and recognize that these data collected within the larger ABCD study will lend themselves to more complicated and statistically sound analyses. As noted above, and in the introduction of our manuscript, this study aimed to establish the acceptability and feasibility of collecting these sorts of data in children. We expect that there will be several studies that build directly on the work we present in this manuscript.

5 . “<$50K/year” USD (Table 1).

We appreciate the reviewer’s close reading.  We have included USD in the header “Annual Household Income”

  1. It is necessary to describe, explain, the most important finding of the study at the beginning of the discussion section.

We appreciate the reviewers comment and have edited the wording of the beginning of the discussion section so as to be clearer what our key findings are (lines 265-273)

  1. Conclusions section 5 is missing. It must be concluded based on the objectives.

Thank you for reminding us of the opportunity to highlight the main message of our manuscript.  We have added Section 5 on lines 333-339.

Reviewer 2 Report

A well written manuscript with only a few points of inquiry:

Was analyzed data normally distributed?  If so by what methodology?

Limitations should also include the major issues related to wrist based: sleep cycle accuracy, HRV and HR during motion for these devices in general. 

Please comment on whether the Fitbit device/data used GPS as part of it’s activity algorithm.  In other words, can we distinguish a participant playing a video game (with vigorous wrist motion) vs playing soccer/running.  Did you consider a strategy where low HR + high actigraphy would be handled separately?

Was the amount/history of exercise training included in the intake questionnaire?  As you are aware, lower resting HR, variable effects on HRV can occur due to training effects.  Resting HR’s below 50 could be common in young athletes (https://onlinelibrary.wiley.com/doi/full/10.1002/clc.23417), causing inappropriate exclusion of data.

Author Response

Authors' Responses:

  1. Was analyzed data normally distributed?  If so by what methodology?

We appreciate the reviewers careful review of our manuscript. For the primary descriptive results, the normality of the distribution of metrics was not critical to our conclusions. For the inferential tests of between group differences, data were visually inspected using histograms, which did not show significant skew.

  1. Limitations should also include the major issues related to wrist based: sleep cycle accuracy, HRV and HR during motion for these devices in general. 

Thank you for this reminder to include the limitations of wrist-based measurement in general, and photoplethysmography in particular. We have included a new paragraph in the limitations portion of the discussion (lines 295-303).

  1. Please comment on whether the Fitbit device/data used GPS as part of it’s activity algorithm.  In other words, can we distinguish a participant playing a video game (with vigorous wrist motion) vs playing soccer/running.  Did you consider a strategy where low HR + high actigraphy would be handled separately?

We appreciate the reviewer’s understanding of the capacity of wearable device’s capacity to incorporate multiple sensors in their algorithms to determine energy expenditure and activity level. The device that was used for this assessment did not include GPS function. This has been made clear in the methods section.

In response to the reviewer’s second point, while we have used actigraphy signal combined with heart rate in other contexts by combining the signal from multiple devices (for instance, BioPac based heart rate assessment and Actigraph based accelerometers) Fitbit does not provide access to the raw accelerometry data. As such, we did not have opportunity to explore this possibility in this context. We did try to address the possibility of “missing” heart rate combined with movement by including minutes that had greater than baseline METS and/or steps (1.0 and 0 respectively) but no heart rate values as legitimate wear time.

  1. Was the amount/history of exercise training included in the intake questionnaire? As you are aware, lower resting HR, variable effects on HRV can occur due to training effects.  Resting HR’s below 50 could be common in young athletes (https://onlinelibrary.wiley.com/doi/full/10.1002/clc.23417), causing inappropriate exclusion of data.

We appreciate the reviewer’s close reading and this point specifically. Unfortunately, we did not include exercise history or training in our intake questionnaire, and we do not have data on the cardiorespiratory fitness of these children. With that said, we included a citation that indicates that resting HR less than 60 is the 10th percentile in this age group, and below 50 is the 1st percentile. Furthermore, the majority of our participants were not highly active and were unlikely to be characterized as athletes.  We expect that this is true of the majority of the youth population (as evidenced in our reference indicating resting HR <50 is the first percentile for this group) and believe that these sorts of very low values are more likely the result of poor wear or artifact than legitimately very low heart rates. 

Round 2

Reviewer 2 Report

Thank you for the clarifications.

Please make sure the abstract editing is correct as I found a couple of typos.

Author Response

Thank you so much for your attention to detail.  All typos have been corrected in the newly uploaded manuscript.